# The Effect of Room Arrangement on the Mood and Milk Volume of Mothers Who Had Cesarean Delivery and Were Not with Their Infants

**DOI:** 10.3390/healthcare12171693

**Published:** 2024-08-25

**Authors:** Nilüfer Ataç, Emine Serap Çağan, Rabia Genç

**Affiliations:** 1Lactation Department, Başakşehir Çam ve Sakura City Hospital, Başakşehir, Istanbul 34480, Turkey; nilufer97atac@gmail.com; 2Midwifery Department, Faculty of Health Sciences, Ağrı İbrahim Çeçen University, Ağrı 04200, Turkey; 3Midwifery Department, Faculty of Health Sciences, Ege University, Izmir 35575, Turkey; rabia.genc@ege.edu.tr

**Keywords:** human milk, human milk expression, emotional state

## Abstract

This study aimed to compare the effect of room arrangement on the mood and milk volume of mothers who gave birth by cesarean section and whose babies were not with them. The study included 89 mothers with and without infants living in the same room (Group 1) and 94 mothers without and without infants living in the same room (Group 2) (n = 183). The expression of milk was performed twice (0–6 h after birth and 36–48 h before discharge) using an electric breast pump. Expression lasted 15 min for each breast separately. The mean first milk volume of the mothers (Group 1: 5.56 ± 5.03 cc, Group 2: 3.76 ± 3.73 cc) and the mean last milk volume (Group 1: 18.78 ± 15.43 cc, Group 2: 10.64 ± 14.12 cc) were determined, and a significant difference was found between the groups. The mean milk volume of Group 1 was found to be significantly higher than Group 2 (Group 1: 13.21 cc ± 14.62 cc, Group 2: 6.88 cc ± 13.56 cc) (*p* < 0.05). A statistically significant difference was found between the two groups in terms of positive and negative mood (*p* = 0.003). In conclusion, it was determined that the milking behavior of mothers was negatively affected due to the lack of stimulation in the room where there was no baby, and this situation negatively affected the start time of the first milking and the average milk volume.

## 1. Introduction

Breast milk is a unique source of nutrition that contains all the nutrients that newborns need from their first moments of life [1]. The World Health Organization (WHO) and the United Nations Children’s Fund (UNICEF) recommend exclusive breastfeeding for the first six months starting immediately after childbirth and breastfeeding and supplementary feeding from the sixth month until the age of two years and beyond. There are also attempts to identify barriers to sustaining breastfeeding [2,3]. Globally, breastfeeding rates and duration are considered a public health priority [4]. However, global breastfeeding rates remain below the target level, despite efforts to promote breastfeeding [5]. Among the factors affecting breastfeeding rates are factors specific to the mother and infant. Factors such as the mode of delivery, the medical condition of the infant, and spouse/partner support can influence the breastfeeding experience [2,6,7]. 

Preterm birth is a serious public health problem at both the national and global levels and refers to childbirth before 37 weeks of gestation [8,9]. Worldwide, around 13.4 million infants were born preterm in 2020. This is one of the leading causes of death in children aged under five and caused approximately 900,000 deaths in 2019 [10,11]. Breast milk, the most suitable food for newborn nutrition, has many benefits for every baby, both term and preterm. It reduces mortality and morbidity rates. Breast milk is fresh, clean, and easy to digest, so it protects against diseases such as diarrhea, pneumonia, type II diabetes, and obesity. The immunological factors found in breast milk play a role in increasing the immunity of the newborn. IgG and IgA found in breast milk bind to foreign protein molecules such as viruses and bacteria and prevent their absorption in the baby’s digestive system. The sodium, potassium, phosphorus, and calcium found in breast milk meet the electrolyte and mineral composition that the baby needs. Allergy-related respiratory problems and food allergies are less common in breastfed babies [12,13,14,15].

Breastfeeding has been described as the only thing mothers can do for their babies when they are in the neonatal intensive care unit, and research has shown that breastfeeding is a unique maternal activity that provides mothers with a sense of purpose and value and helps them feel a bond and connection with their babies [16]. Mothers of infants hospitalized in the neonatal intensive care unit (NICU) may experience difficulties in breastfeeding due to various newborn- and NICU-related obstacles [17]. It is important to initiate milking within the first six hours after delivery and encourage breastfeeding [18,19]. Premature infants’ stays in the NICU may have negative effects on mothers [20,21]. Mothers whose infants are in the NICU may experience emotional distress, which may negatively affect their milk production [21,22]. Maternal stress can reduce milk production by disrupting oxytocin release in the mother, disrupting milk synthesis in lactocytes, reducing insulin sensitivity and secretion, and causing HPA axis irregularity, which can negatively affect milk production and secretion [23]. This situation can create an additional source of stress for mothers [21,22]. The benefits of expressing breast milk for babies and mothers have been identified. However, breast milk expression rates were found to be quite low according to a study conducted in the United States (US), while no statistical data could be found on breast milk expression in Turkey [21,24].

Despite studies on the content and quantity of breast milk and breastfeeding behaviors, there is not sufficient information on the relationship between the mood and milk volume of mothers who stay in the room while their infants are in the NICU. It is also not clear how the intention and motivation to express breast milk are affected in hospital rooms where there is some kind of peer interaction. There is no research on whether there is a relationship between the emotional state and milk volume of mothers who stay in the same room since birth and the bond they establish with other mothers who share their first experiences. Therefore, this study aimed to examine the effect of a room arrangement for mothers who had cesarean delivery and who were not with their infants on their mood and milk volume.

## 2. Materials and Methods

### 2.1. Research Design

This study was a comparative cross-sectional study. 

### 2.2. Setting and Relevant Context

The scope of this study consisted of mothers who were hospitalized and gave birth in the postpartum wards of Başakşehir Çam and Sakura City Hospital in Istanbul, Turkey, between February 2023 and April 2023. The hospital is a baby-friendly hospital and has five postpartum wards. Each ward consists of 24 beds and the total capacity is 120 beds. Although our hospital is baby-friendly, mothers’ visits to their babies are limited within the scope of the “neonatal intensive care unit infection control procedure” and mothers can visit at certain hours and during breastfeeding processes.

### 2.3. Sample

The study sample consisted of mothers who were hospitalized in the postpartum wards of Başakşehir Çam and Sakura City Hospital in the specified period, who gave birth, and who met the inclusion criteria of the study (n = 183). The study was conducted with two different groups: 89 mothers who were not with their infants and were staying in the same room with mothers who were with their infants (Group 1) and 94 mothers who were not with their infants and were staying in the same room with mothers who were not with their infants (Group 2) (Figure 1).

Inclusion criteria: The inclusion criteria of the study were mothers giving birth within the 32nd−37th weeks of gestation, having a cesarean section, having an infant hospitalized in the neonatal intensive care unit, staying in a double room, and volunteering to participate in the study.

Exclusion criteria: Mothers who were aged under 18, who gave birth at less than 32 weeks and over 37 gestational weeks, who could not read/write in Turkish, who had a normal vaginal delivery, who had a stillbirth, who had a mastectomy, who were with their infants, who stayed in a single room, who were diagnosed with or at risk of postpartum depression or maternal blues, who were receiving treatment, and who did not want to participate in the study were excluded from this study.

### 2.4. Measurement

In terms of room arrangement, mothers were randomly placed in double rooms without any arrangement. Then, mothers staying in double rooms were evaluated according to the details of the mothers sharing their rooms with their babies. No intervention was made by the researcher regarding the room arrangement. Mothers were placed in rooms according to availability. Then, the mothers who met the inclusion criteria and volunteered to participate in the study filled out the “Informed Voluntary Consent Form”, the “Individual Introduction Form”, and the “Mood Assessment Scale”, and it was ensured that they expressed milk. It took approximately 10−15 min to administer the survey to the mothers, followed by approximately 30 min of milking.

Expressing breast milk and the assessment of milk volume were studied as follows: In this study, milking was performed twice. The amount of milk in the first 0−6 h and the amount of milk before discharge (36−48 h postpartum) were measured in the mothers who were not with their infants during the study. The amount of milk was measured 3 h after the last milking by milking both breasts for 15 min and then monitoring the amount expressed. To express breast milk, the only type of free-standing electric breast pump of the hospital, “Babyvacc TBM3100” (Babyvacc, İzmir, Turkey), was used. Milking devices are regularly calibrated once a year to maintain accurate and reliable functioning. This is an important routine to check their performance and ensure their long-term use. Injectors were used to measure the amount of milk expressed. The injector size (5, 10, or 20) was chosen according to the amount of milk expressed. Lactation stimulation was not applied to mothers for expressing milk.

### 2.5. Data Collection 

An “Individual Introduction Form” developed by the researchers based on the literature and the “Mood Assessment Scale” were used for data collection.

Individual Introduction Form: This form consists of a total of 13 questions regarding the sociodemographic characteristics of the mothers. Questions include the mother’s age, education and income status, number of births, week of birth, admission to the neonatal intensive care unit in the previous pregnancy, reason for cesarean section, gender of the baby, etc. The questions were prepared in line with the literature [3,24,25,26].

Mood Assessment Scale: The original version of this scale was developed by Akdoğan for school administrators, and its validity and reliability studies were conducted by Yıldırım and Tabak [27,28]. It was developed to describe the emotional states of teacher candidates studying at a university in Turkey with different classifications [28]. The scale determines the frequency and intensity of 54 different emotions. It includes items for positive and negative emotions. There are two subdimensions in the scale: positive emotions and negative emotions. The positive emotions subdimension includes items such as pride, encouragement, enthusiasm, peace, compassion, gratitude, love, and hope. The negative emotions subdimension includes items such as boredom, helplessness, disappointment, anxiety, grief, unhappiness, hatred, anger, loss of hope, and loneliness. The frequency of experiencing the emotion is rated on a five point Likert type scale (1 = Never, 5 = Always), whereas the intensity of the emotion is rated on a three point Likert type scale (1 = Low, 2 = Moderate, 3 = High). Mood is calculated by multiplying the frequency and intensity of the emotion. The evaluation of the mood score is also based on a five point Likert type scale (1 = Almost none, 5 = Extremely). In Yıldırım and Tabak’s study, it was determined that the general Cronbach Alpha value of the scale was 0.82, and the Cronbach Alpha value of its sub-dimensions was between 0.80 and 0.85 [28].

### 2.6. Data Analysis

In the analysis of the data, the IBM SPSS Statistics 26.0 package program (SPSS Inc., Chicago, IL, USA) was used. Descriptive statistics were calculated with numbers and percentages, means were shown with standard deviations, and *p* < 0.05 was accepted as the statistical significance level. In the evaluation of the relationships between dependent and independent variables, the chi-square test was used for categorical data and the *t*-test and Mann–Whitney U were used for continuous variables.

### 2.7. Ethical Approval

Ethics committee approval was obtained from the Ege University Medical Research Ethics Committee (Date: 23 February 2023, Decision No: 23-2.1T/21). To conduct the study in Başakşehir Çam and Sakura City Hospital, permission was obtained from the Istanbul Provincial Directorate of Health (Date: 25 April 2023, Decision No: 2023/06). Mothers who met the inclusion criteria and agreed to participate in the study were informed that their personal information would not be used for any purpose other than this research. Written informed consent was obtained from the participants. The study was carried out in accordance with the principles of the Declaration of Helsinki (https://www.wma.net/what-wedo/medical-ethics/declaration-of-helsinki/, accessed on 24 February 2023).

## 3. Results

The mean age of the mothers who participated in the study was 29.66 ± 5.94 in Group 1 and 30.85 ± 6.36 in Group 2. All women had a cesarean section with epidural anesthesia. None of the women had a chronic health problem (such as gestational diabetes, hypothyroidism, or obesity). The indication for cesarean section in 67% of mothers in Group 1 and 71.9% of mothers in Group 2 was pregnancy-related problems (such as pre-eclampsia, placental anomalies, multiple pregnancy, and fetal distress). There were no women breastfeeding in tandem. Table 1 shows the findings regarding the sociodemographic characteristics of the mothers who participated in the study.

According to the findings regarding the time and frequency of milking (Table 2), the mean time of milking of the mothers was different in Group 1 and Group 2, and the mean of Group 2 (7.57 ± 4.39) was significantly higher than that of Group 1 (6.82 ± 3.66) (*p* < 0.05). The majority of the mothers in Group 1 (n = 63, 67.0%) had a frequency of milking every three hours, and the majority of the mothers in Group 2 (n = 46, 51.7%) had a frequency of every three hours. There was no significant correlation between the frequency of milking of both groups (*p* > 0.05).

When the first and last milk volumes of the mothers in Group 1 and Group 2 were compared, a significant difference was found between the mean first and last milk volumes and mean milk volume differences in the mothers in both groups, and the mean milk volume of the mothers in Group 1 (13.21 ± 14.62) was significantly higher than that of the mothers in Group 2 (6.88 ± 13.56) (*p* < 0.05) (Table 3).

When the mean scores of the mothers in the groups on the mood assessment scale were evaluated, a statistically significant difference was found between the groups in terms of the frequency of positive emotions, frequency of negative emotions, intensity of positive and negative emotions, and positive and negative mood (*p* < 0.05) (Table 4).

## 4. Discussion

Mothers of preterm infants with low birth weight often have difficulty succeeding at breastfeeding and are more likely to discontinue breastfeeding than mothers of term infants. Thus, infants who should benefit the most from breast milk are often not able to benefit from it. One possible solution to the barriers to breastfeeding for infants at risk is the earlier initiation of milking as a method to increase breast milk production [29,30,31,32]. It was determined that the average milking start time in Group 2 was significantly higher than in Group 1 (*p* < 0.05). This may be attributed to the fact that mothers in Group 1 may have been affected by the sound or sight of the infants of the mothers they shared the room with and may have tended to start milking earlier, thinking of their infants. In the analysis of the frequency of milking, it was determined that milking every three hours was the most common finding in both groups, and no significant correlation was found between the groups in this regard (*p* > 0.05). When the first and last milk volumes were analyzed between the groups, significant differences were found between the groups in terms of the first and last milk volumes, and it was determined that the mothers in Group 1 had higher milk volumes at the baseline and at the last measurement (*p* < 0.001). Likewise, significant differences were found between the first and last milk volumes of the mothers in Group 2 (*p* < 0.001). In both groups, the first milk volume of the mothers was higher than their last milk volume. The mothers in Group 1 had a higher milk volume than those in Group 2. This may be because the time of starting milking was earlier in Group 1 than in Group 2. 

Mothers who are not with their infants often face several challenges in delivering postpartum breast stimulation. Despite these challenges, early milking should be a priority to optimize the lactation outcomes of mothers [33]. Furman et al. found that mothers who started milking within the first 6 h after delivery were more likely to breastfeed at the 40th adjusted gestational week [34]. Similar to the study of Furman et al., Parker et al. found that mothers who started milking within six hours produced more breast milk during the first milking session and on the 6th, 7th, and 42nd days [32]. In the study conducted by Parker et al. with 20 mothers who gave birth to very-low-birth-weight infants, it was reported that starting milking within 60 min was associated with increased breast milk production for 6 weeks [35]. Contrary to other studies and our study, Parker et al. reported that starting milking between the 181st and 360th minutes after delivery resulted in an increase in milk production for six weeks postpartum [36]. The findings obtained in our study are consistent with the results of the studies conducted by Furman et al. and Parker et al., which showed that the early onset of milking affects milk volume [34,35,36]. Although studies generally state the first six hours in terms of milk expression time, there are also studies that specify different time intervals. The reason for this is thought to be related to factors such as a small sample size, milk expression initiation periods longer than 24 h after birth, the general condition of the mother and baby, the week of birth, the psychology of the mother, and the delivery method [32,36]. In our study, it was aimed to ensure reliability by standardizing the time and frequency of milking and the method of delivery.

Emotions are the forces that enable an individual to act on intrinsic impulses. Depending on stimuli from the internal and external environment, people develop emotional reactions to other people, objects, or events in and around themselves [28]. Women with negative emotions and psychological distress (perceived stress, anxiety, depression, etc.), who are stressed and have reduced attachment to their newborn, have lower breastfeeding markers and are more likely to experience breastfeeding difficulties and discontinue breastfeeding [23,37]. In our study, mothers in Group 1 experienced positive emotions more frequently; meanwhile, mothers in Group 2 had lower mean scores on this issue, and mothers in Group 2 expressed negative emotions more frequently. When the differences in the moods of the mothers in the groups were associated with their milking habits, it was found that it took significantly more time for the mothers in Group 2 to start milking compared to the mothers in Group 1. This suggests that there may be a connection between mood and milking habits. Although no significant difference was found in the frequency of milking in both groups, it is noteworthy that the mothers in Group 2 started milking later. Based on the findings of our study, it is thought that the presence of another infant in the room may affect the mood and milking habits of the mothers evaluated in the groups. The presence of an infant in the room in which the mothers in Group 1 stayed and the absence of an infant in the room in which the mothers in Group 2 stayed emphasize this finding more. In particular, it was found that the mothers in Group 1 had a more positive mood and started milking earlier. This suggests that the presence of another infant in the room may positively affect the mother’s mood, and this positive effect was reflected in their milking habits. Regarding mood, the mothers in Group 2 had higher negative emotion scores, indicating that the absence of an infant in the room may cause emotional difficulties and a lack of motivation in mothers. It is thought that this may negatively affect milking habits and affect the time when milking is started and the amount of milk produced. Similar to our findings, Alves et al. (2016) reported that the main reasons for breastfeeding barriers in mothers who were separated from their infants were concerns about insufficient milk supply (35.7%), difficulties in expressing milk (24.8%), and physical separation from their infants (24.3%) [38]. There is a limited number of studies in which the relationship between mothers’ moods, the amount of milking, and the presence of an infant have been examined. In a few studies in the literature, it has been shown that social barriers are stressors that may impede the start of breastfeeding. Studies have focused more on moods such as anxiety, stress, and depression. Foligno et al. (2020) reported that mothers with high stress levels had reduced breastfeeding rates during their hospital stay [39]. In addition, Caparros-Gonzalez et al. (2019) reported that high cortisol levels reduce breast milk production [40]. Similar to other studies, the findings obtained in our study show that mothers’ moods affect the time of milking and milk volume. 

### Limitations 

Loss of follow-up in the collection of follow-up data of this study and only examining mothers who gave birth prematurely are the limitations of the study. In addition, it is among the limitations of the study that the postpartum women sharing the room are not placed in the room at the same time and are not discharged at the same time, and the postpartum mothers next to them change according to the discharge time of the postpartum women with and without a baby when they are first admitted to the hospital. The support of family members was not evaluated during the study. This is also among the limitations. Additionally, the fact that the study was conducted in a single hospital is among the limitations. 

## 5. Conclusions

As a result, it seems that the presence of another baby in the room is an important factor in mothers’ emotional states and milking habits. These findings suggest that the milking process may be affected by emotional and environmental factors in a complex way, and these interactions require further investigation. This study aims to examine the effects of maternal emotional states on milking behaviors in more depth and to evaluate emotional state measurements more specifically. The positive effects of starting early milk expression on milk production should be emphasized more, and mothers should be made aware of this issue. The factors affecting breast milk expression habits and emotional states should be examined in more detail (including family support, social support, healthcare personnel approach, etc.) should babies be transferred to the NICU. Considering that mothers who are in the hospital are affected by the presence of another baby in their room, more research should be conducted to understand the effects on their milking habits and emotional states, and guidance services should be provided by health professionals to mothers who are not with their babies, especially to increase their motivation and emotional support during the milking process.

## Figures and Tables

**Figure 1 healthcare-12-01693-f001:**
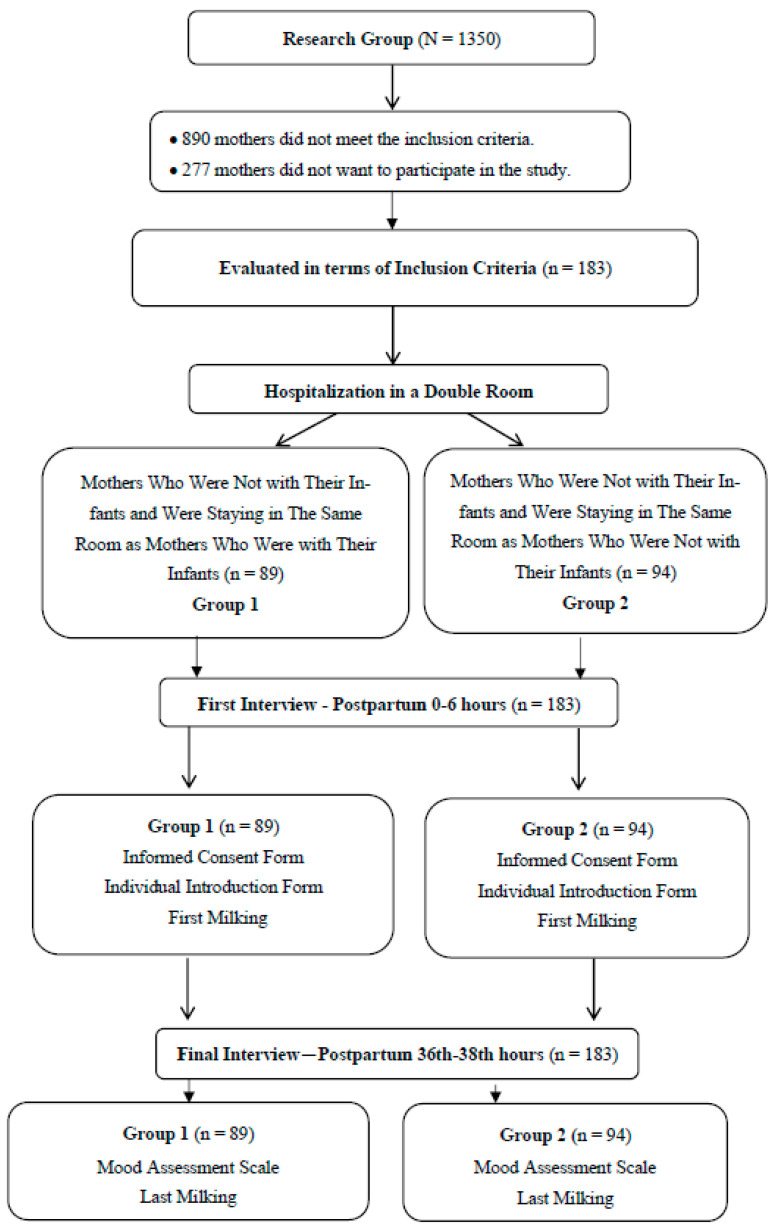
Research Flowchart.

**Table 1 healthcare-12-01693-t001:** Distribution of Sociodemographic Characteristics of Mothers.

Variables	Groups				Statistical Value	
	Group 1		Group 2			
	Mean ± SD. (Min.−Max.)		Mean ± SD. (Min.−Max.)		X^2^/U	*p*
Age	29.66 ± 5.94 (19−43)		30.85 ± 6.36 (19−44)		−1.248 ^b^	0.212
Gravida	2.50 ± 1.78 (1−15)		2.49 ± 1.46 (1−10)		−0.287 ^b^	0.774
Parity	2.18 ± 1.04 (1−5)		2.30 ± 1.26 (1−7)		−0.395 ^b^	0.693
Gestational Week	35.02 ± 1.23 (31−37)		34.70 ± 1.33 (32−39)		−2.047 ^b^	0.041 *
Variables	N	%	n	%	X^2^/U	*p*
Marital Status						
Married	93	98.9	89	100.0	0.952 ^a^	0.329
Single	1	1.1	0	0.0		
Duration of Marriage						
Mean ± SD (Min.–Max.)	7.32 ± 5.39 (0−20)		8.57 ± 5.88 (1−23)		−1.390 ^b^	0.165
Income Status						
Income less than expenses	11	11.7	15	16.9	5.034 ^a^	0.081
Income equal to expenses	80	85.1	65	73.0		
Income more than expenses	3	3.2	9	10.1		
Indication for Cesarean Section						
Repeated Cesarean section	31	33.0	25	28.1	3.230 ^a^	0.919
Fetal Distress	32	34.0	31	34.8		
Pre-eclampsia	13	13.8	15	16.9		
Multiple Pregnancy	8	8.5	6	6.7		
Placental Anomalies	6	6.4	7	7.9		
Other Reasons	4	4.3	5	5.6		
Baby’s Gender						
Girl	37	39.4	34	38.2	0.580 ^a^	0.446
Boy	57	60.6	55	61.8		
Total	94	100.0	89	100.0		

* *p* < 0.05, Test Used: a: Chi-square Test, b: Mann–Whitney U.

**Table 2 healthcare-12-01693-t002:** Findings regarding the time and frequency of milking.

Variables	Groups				Statistical Value	
	Group 1		Group 2			
	Mean ± SD		Mean ± SD		X^2^/U	*p*
Time of Milking	6.82 ± 3.66 (2–24)		7.57 ± 4.39 (1–26)		−2.167 ^b^	0.030 *
Frequency of Milking	N	%	n	%	X^2^/U	*p*
Once an hour	4	4.3	1	1.1	8.578 ^a^	0.073
Once every 2 h	7	7.4	12	13.5		
Once every 3 h	63	67.0	46	51.7		
Once every 4 h	12	12.8	22	24.7		
Once every 5 h	8	8.5	8	9.0		
Total	94	100.0	89	100.0		

* *p* < 0.05, Test Used: a: Chi-square Test, b: Mann–Whitney U.

**Table 3 healthcare-12-01693-t003:** Comparison of milk volume of groups in the first and last milking.

Milking Characteristics	Group 1 (n = 94)	Group 2 (n = 89)	Statistical Value	
	Mean ± SD	Mean ± SD	U	*p*
Milk Volume at First Milking (cc)	5.56± 5.03	3.76± 3.73	−2.624 ^b^	0.009 **
Milk Volume at Last Milking (cc)	18.78± 15.43	10.64 ±14.12	−5.094 ^b^	<0.001 ***
(Last–First) Milking Volume (cc)	13.21± 14.62	6.88 ±13.56	−4.878 ^b^	<0.001 ***

*** *p* < 0.001, ** *p* < 0.01, Test Used: b: Mann Whitney U.

**Table 4 healthcare-12-01693-t004:** Comparison of mood assessment scale scores of mothers in groups.

Variables of the Mood Assessment Scale	Group 1 (n = 94)	Group 2 (n = 89)	Statistical Value	
	Mean ± SD	Mean ± SD	t/U	*p*
Positive Emotions (Frequency)	3.06 ± 0.98	2.59 ± 1.15	2.973 ^d^	0.003 *
Negative Emotions (Frequency)	1.99 ± 0.70	2.27 ± 0.85	−2.090 ^b^	0.037 *
Positive Emotions (Frequency)	2.00 ± 0.61	1.81 ± 0.56	−2.148 ^b^	0.032 *
Negative Emotions (Frequency)	1.54 ± 0.43	1.70 ± 0.49	−2.063 ^b^	0.039 *
Positive Mood	2.16 ± 1.14	1.68 ± 1.09	−2.980 ^b^	0.003 *
Negative Mood	1.09 ± 0.62	1.37 ± 0.86	−2.350 ^b^	0.019 *

* *p* < 0.05. Test Used: b: Mann–Whitney U, d: Independent Samples *t*-Test.

## Data Availability

The raw data supporting the conclusions of this article will be made available by the authors upon reasonable request after signing a confidentiality agreement.

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
