# Peer review of "The Effect of Room Arrangement on the Mood and Milk Volume of Mothers Who Had Cesarean Delivery and Were Not with Their Infants"

_healthcare, 2024, doi:10.3390/healthcare12171693_

Round 1

Reviewer 1 Report

Comments and Suggestions for Authors

The summary in the method section does not provide precise information on which day after the cesarean section the patients were examined. Was the milk expressed with a breast pump or manually? If with a breast pump, which one? Single-phase or two-phase? On one breast or both at the same time?

Line 46:

Please elaborate on why experiencing emotional stress can negatively affect milk production?

Were anxiety, depression, baby blues episodes included in the inclusion/exclusion criteria for the study? Treatment for depression? The occurrence of risk factors for postpartum depression?

Did the patients suffer from gestational diabetes, obesity, thyroid disease? - these patients may have delayed milk supply and therefore need more time to produce larger volumes of milk.

Line 148: All women had no maternal health problems. while in Table 1 the indications for performing a cesarean section Preeclampsia - how to interpret it?

Did the patients stimulate lactation in any way?

Why wasn't a parameter such as milk cortisol/cortisol in blood serum taken into account, it would certainly be related to mood and possibly to the volume of milk extracted.

And was a factor such as support from the woman's family? friends? medical personnel taken into account during the study?

The conclusions are a repetition of the results. Please edit the entire part referring to the conclusions.

Comments on the Quality of English Language

 Minor editing of English language required.

Author Response

Thank you very much for taking the time to review this manuscript. Explanations regarding the edits we made based on your suggestions are included in the attached document.

Kind regards

Reviewer 2 Report

Comments and Suggestions for Authors

Abstract: The abstract lacks clarity

1-L 14-15: Clearly define Group 1 and Group 2

2- L 17-18: Provide the unit of measurement

3- The Results do not support the conclusion. For example, there is no mention of the first milking in the Results – instead, only an average was reported

Introduction: The Introduction is not well-focused.

1-    The first paragraph is about preterm birth, although there is no mention of preterm delivery in the title of the study or the abstract

2-    The authors should discuss the benefits of breastmilk for preterm infants

Materials and Methods:

1-    L 57-58: State which country

2-    L 58-61: Provide more information on the reasons the mother's access to the infant is restricted, although the hospital is designated a Baby-Friendly Hospital setting

3-    LL 74-75: Exclusion criteria: What about weeks of gestation at delivery?

4-    Fig. 1: Add: How many women were invited, how many refused, and for what reasons

5-    LL 117-120: Was the questionnaire pretested? Provide information about the development of the questionnaire

6-    L 121-122: Akbulut (2016) is missing from the References section. Use a consistent citation format throughout the manuscript (author/year or numbering)

7-    L128-129:  Provide more details about the Mood questionnaire. What population and country was it initially developed for and used?

8-    Clarify the cited reliability score of the questionnaire. Is it that of the original questionnaire or based on the current study?

9-    L 138-146: There is no mention of obtaining informed consent from participants

Results:

Tables 1 and 2: Fix the alignment of the variables and the reporting unit. For example, age is reported as Mean and SD but placed under the subheading N, %

Discussion

L 186-188: Add citations to support the statements

LL 190-205 and 230-239:  Discuss the Results of the study supporting with the relevant literature instead of repeating the results

Limitations: The study was limited to only one hospital. The authors should acknowledge this as one of the limitations.

The authors also need to acknowledge the existence of other potential reasons that could affect milking and state this limitation as part of the conclusions

Author Response

(The authors gave the same response as above.)

Reviewer 3 Report

Comments and Suggestions for Authors

The article submitted to Healthcare by Emine Serap ÇaÄŸan et al. is devoted to studying two parameters - mood and milk volume - in mothers who gave birth by cesarean section and were kept separate from their infants.

In the Abstract section, the reviewer suggests to add information about how the two groups of women differed (from lines 62-69)

In the Introduction section, some literature data confirming the rationality of the proposed experimental design and formulating the hypothesis that the authors tried to test in the article is recommended

In the Discussion section, the authors should discuss the results obtained and the reliability of the differences from a physiological point of view.

Comments on the Quality of English Language

It is recommended that a professional scientific editor edit the text.

Author Response

Thank you very much for taking the time to review this manuscript. Explanations regarding the edits we made based on your suggestions are included in the attached document.We hope we understood your suggestion correctly and were able to make edits and additions.

Kind regards

Round 2

Reviewer 1 Report

Comments and Suggestions for Authors

Thank you for your detailed answers.

In your next planned studies, please include laboratory results - the manuscript will be more interesting.

Author Response

Thank you very much for taking the time to review this manuscript. An interventional study is currently being planned to evaluate hormones.

Reviewer 2 Report

Comments and Suggestions for Authors

The authors addressed the comments; however, the validity of a mood questionnaire [28] designed for school administrators (i.e., unrelated to maternal postpartum mood) is unclear. Moreover, since the original study is available only in Turkish, the authors should explain the relevance of the items in the original questionnaire and provide the English translation of the items in the scale for the readers. 

Author Response

Thank you very much for taking the time to review this manuscript. The emotional state scale was developed for teachers, but the scale items do not include items for a specific group of teachers. It includes items for positive and negative emotions. There are two sub-dimensions in the scale: Positive emotions and negative emotions. The positive emotions sub-dimension includes items such as pride, encouragement, enthusiasm, peace, compassion, gratitude, love, and hope. The negative emotions sub-dimension includes items such as boredom, helplessness, disappointment, anxiety, grief, unhappiness, hatred, anger, loss of hope, and loneliness. The frequency of experiencing emotions is evaluated on a five-point Likert-type scale, while the intensity of experiencing emotions is evaluated on a three-point Likert-type scale. The frequency of experiencing the emotion is rated on a five-point Likert-type scale (1: Never - 5: Always), whereas the intensity of the emotion is rated on a three-point Likert-type scale (1: Low, 2: Moderate, 3: High). Mood is calculated by multiplying the frequency and intensity of the emotion. The evaluation of the mood score is also based on a five-point Likert-type scale (1: Almost none, 5: Extremely). When permission to use the scale was sought, information about the study group and the name of the study was provided, and approval for use was obtained from the scale author. The English version of the scale is included in the appendix.

Reviewer 3 Report

Comments and Suggestions for Authors

The manuscript has been significantly improved according to the reviewers' comments.

Lines 136-137: grup -> group?

Author Response

Thank you very much for taking the time to review this manuscript. We changed to Group from Grup in Figure 1.
